# Endophytic Fungi from *Terminalia* Species: A Comprehensive Review

**DOI:** 10.3390/jof5020043

**Published:** 2019-05-24

**Authors:** Rufin Marie Kouipou Toghueo, Fabrice Fekam Boyom

**Affiliations:** Antimicrobial and Biocontrol Agents Unit (AmBcAU), Laboratory for Phytobiochemistry and Medicinal Plants Studies, Department of Biochemistry, Faculty of Science, University of Yaoundé I, P.O. Box 812, Yaoundé, Cameroon; fabrice.boyom@fulbrightmail.org

**Keywords:** *Terminalia* species, endophytic fungi, diversity, biological activities, secondary metabolites

## Abstract

Endophytic fungi have proven their usefulness for drug discovery, as suggested by the structural complexity and chemical diversity of their secondary metabolites. The diversity and biological activities of endophytic fungi from the *Terminalia* species have been reported. Therefore, we set out to discuss the influence of seasons, locations, and even the plant species on the diversity of endophytic fungi, as well as their biological activities and secondary metabolites isolated from potent strains. Our investigation reveals that among the 200–250 *Terminalia* species reported, only thirteen species have been studied so far for their endophytic fungi content. Overall, more than 47 fungi genera have been reported from the *Terminalia* species, and metabolites produced by some of these fungi exhibited diverse biological activities including antimicrobial, antioxidant, antimalarial, anti-inflammatory, anti-hypercholesterolemic, anticancer, and biocontrol varieties. Moreover, more than 40 compounds with eighteen newly described secondary metabolites were reported; among these, metabolites are the well-known anticancer drugs, a group that includes taxol, antioxidant compounds, isopestacin, and pestacin. This summary of data illustrates the considerable diversity and biological potential of fungal endophytes of the *Terminalia* species and gives insight into important findings while paving the way for future investigations.

## 1. Introduction

Instead of living alone as single entities, plants are closely associated with microorganisms present in their neighborhood and, particularly, their microbiome [1]. Consequently, microbes are an essential part of the phenotype, influencing the fitness and, thus, the ecologically important traits of their hosts [2]. Indeed, plants host communities of microbes called endophytes that influence a wide variety of their biological processes [3]. These endophytes mainly consist of bacteria and fungi that colonize and spend the whole or part of their life cycles inside the plant tissues without instigating any noticeable symptoms of infection or visible manifestation of diseases to their hosts [4]. It appears that endophytes are ubiquitous in nature, as they have been isolated from almost every plant species examined [5]. Of note, of the nearly 300,000 plant species forming the vegetal biodiversity of earth, each individual plant is host to one or more endophytes and can consequently constitutes an opportunity to find new and interesting endophytic microorganisms [6]. In fact, until now, only few of the existing plants have been completely studied in relation to their endophytic content [7]. Therefore, there is a need to expand the investigation to other plant species. 

The initial step in dealing with endophytic microorganisms is the selection of a proper and promising plant species. The reasons behind host plant selection include to investigate plants that are used in traditional medicine for the treatment of infections and to identify the endophytes found in different parts of those plants [8]. Indeed, plants with ethnobotanical history are good candidates for endophytes study since the medical uses for which the plant may have been selected relate more to its population of endophytes than to the plant biochemistry itself [6,7,8]. Moreover, the discovery that some endophytes living inside the host plant can produce the same compounds as the host [9] led to the assumption that the healing processes, as discovered by indigenous peoples, might be facilitated by compounds produced by one or more specific plant-associated endophytes, as well as the plant products themselves [10].

Plants belonging to the genus *Terminalia* (Combretaceae family) are used for traditional medicinal purposes worldwide. The genus *Terminalia* comprises approximately 200–250 species of medium-to-large flowering trees, most reaching up to 75 m tall in height [11]. Members of the genus *Terminalia* have a long history in traditional medicinal systems since they are widely used in several continents for the treatment of numerous diseases, including cardiovascular effects [12], wound healing [13], abdominal disorders, bacterial infections, colds, sore throats, conjunctivitis, diarrhoea, dysentery, fever, gastric ulcers, headaches, heart diseases, hookworm, hypertension, jaundice, leprosy, nosebleed, edema, pneumonia and skin diseases [14,15]. Apart from their ethnopharmacological usage, several pharmacological properties, including antibacterial [16,17], antifungal [18], antiprotozoal [19,20], antiviral [21,22], antidiarrhoeal [23], analgesic [24], antimalarial [25,26], antioxidant [27,28], anti-inflammatory [29,30,31], antidiabetic [32,33], antihyperlipidemic [34,35], anticancer [36,37,38], and anti-HIV [25,39] activities—as well as several bioactive compounds—have been reported from some of these species [17]. However, even though several authors have reviewed the phytochemistry, pharmacological activities, and ethnopharmacological usage of the *Terminalia* species, no review on the *Terminalia’s* endophytic fungi and their bioactive compounds has been reported, to the best of our knowledge. Therefore, in the following text, we reviewed the distribution of endophytic fungi in organs of *Terminalia* spp., the reported bioactive potential of crude extracts, and the purified substances obtained from different endophytic fungi.

## 2. Biodiversity of Endophytic Fungi Isolated from Different *Terminalia* Species

Fungi are one of the most diverse life forms on this planet. After decades of research on fungal endophytes, it is now clear that they are unexceptionally present in all taxonomic groups of the plant kingdom, vegetation types (alpine to tropical), and ecological types (hydrophytes to xerophytes) in great diversity [40,41]. According to Arnold et al. [42], fungal endophytes are hyperdiverse in the tropics. Therefore, we may expect that *Terminalia* spp., which is mostly distributed in tropical regions of the world, could be the host of highly diverse endophytic fungi communities. Table 1 summarizes the distribution and diversity of fungi genera reported from *Terminalia* spp. over the past fourteen years.

### 2.1. Diversity of Fungal Genera Isolated from Terminalia spp. as a Function of Locations and Seasons

The diversity and frequency of the endophytic fungi population have been reported to be greatly affected by the climatic conditions and the location of where the host plant grows [61,62]. The analysis of the table shows a difference in endophyte colonization among various sampling sites. In fact, the fungi of inner bark and twig samples of five *T. arjuna* specimens collected at three riparian locations in India were studied by Tejesvi et al. [51]. The genera *Pestalotiopsis* (54.5%), *Chaetomium* (10.5%), and *Myrothecium* (9%) were the most predominant endophytes. However, some of the fungi were apparently restricted or showed some preference for a particular location. *Pestalotiopsis* was isolated from almost all the locations investigated, while *Tubercularia vulgaris* was restricted to Srirangapatna. In addition, Tejesvi et al. [49] reported that for endophytic fungi colonizing inner bark of *T. arjuna* collected in Riparian zone of India, a great diversity was observed. Fungi belonging to *Chaetomium*, *Botryosdiplodia*, *Pestaliotiopsis*, and *Trichoderma* were isolated from the three regions investigated. However, *Cladosporium*, *Cochlonema*, *Humicola*, *Myrothecium*, *Nigrospora*, *Phialophora*, and *Stemphylium* were isolated only in the Mysore region. *Gliocladium* and *Memnoniella* were isolated only in samples from the Nanjangud region, while *Chloridium*, *Myrothecium*, *Monocillium*, and *Tubercularia* were specific to the Srirangapatra region.

Tejesvi et al. [59] also reported a great variability in endophytic fungi content in twigs samples depending on the region where *T. arjuna* was sampled. In fact, twigs collected from the Mysore region were dominated only by *Pestalotiopsis microspore*, while those collected in the Nanjangud region were dominated by *Pestalotiopsis theae.* Womersley [60] also reported the isolation of *Pestalotiopsis microspora* from the stem of *T. morobensis* collected in Papua New Guinea. Phaopongthai et al. [44] reported that the leaves of *T. chebula* collected from Suanluang Rama IX Public Park, Bangkok (Thailand) were colonized by *Alternaria alternate.* Krittapong et al. [55] reported the presence of *Glomerella* sp., *Paecilomyces* sp., *Phomopsis* sp., *Phyllosticta* spp., and mycelia sterilia in the leaves of *T. chebula* collected in the deciduous dipterocarp forest in the Tak province of Thailand without any sign of *Alternaria* spp. The bark of *T.chebula* collected in the Gopalaswamy hills were found to be dominated by *Pestalotiopsis*, as reported by Tejesvi et al. [49,59], while the *T. chebula* collected in Dhaka, Bangladesh was dominated by *Penicillium thiomii* [58]. Mookherjee et al. [56] isolated the endophytic fungi *Mycosphaerella crystalline* and *Kwoniella* sp. from the fruit of *T. bellirica* collected at Kharagpur (India), while *Acremonium sclerotigenum* was isolated from the leaves of *T. bellirica* collected from the Bhadrachalam forests (India) [43]. *Oidium* sp. was isolated in *T. catappa* from Costa Rica [57] and was not was isolated from tissues of *T. catappa* growing in Cameroon [50]. This analysis of collected data supports the claim that the location of the host plant has a great influence on the diversity of fungi colonizing plants. In fact, this location-specific distribution of certain fungi genera observed with *Terminalia’s* endophytes is supported by previous reports [63,64]. 

Seasons have also been reported to impact the endophytic fungi population in plants. In fact, Tejesvi et al. [51] found only five species in winter compared to 19 species during monsoon in the bark of *T. arjuna.* Indeed, Mishra et al. [65] reported that maximum fungal diversity was usually observed in the rainy season, as high humidity and temperature favoured the growth of endophytic fungi and also aided the dispersal of spores. In addition, Sadeghi et al. [66] showed that a lower annual rainfall and a comparatively low annual temperature may affect colonization of host tissues by endophytes. According to Krittapong et al. [55], a great variability of composition was found with fungi from the leaves of *T. chebula* collected at two different seasons. In fact, *Paecilomyces* sp. was the most abundant in the dry season, while *Phomopsis* spp. and mycelia sterilia were abundant during the wet season. This frequency of colonization associated with season was previously reported [67]. The change of endophytic fungi colonization can be explained by the fact that fungal symbionts can benefit plants by ameliorating the abiotic stressors associated with climate change such as heat and drought [68,69]. Moreover, the comparison of endophytes mycobiota of the bark of *T. arjuna* collected in Riparian in 2005 and 2006 by Tejesvi and collaborators also suggests that time may also affect the composition of fungal community of plant species.

### 2.2. Influence of Plant Species and Plant Organs on Fungi Colonization

In addition to the locations and seasons, it should be expected that different plant species or organs of plants support different communities and/or levels of richness and abundance. It is well known that diverse endophytes colonize the internal tissues of plants. The analyses of data collected clearly indicate that the endophytic fungi community in *Terminalia* spp. differed from one species to another and from one organ to another (Table 1). In fact, endophytic mycoflora of the mature leaves of *T. alata*, *T. arjuna*, *T. catappa, T. chebula*, and *T. crenulata* sampled in three locations (Kargudi, Ronohills, and Masinagudi) in India were investigated, and six dominant genera were identified; *Phomopsis* and *Colletotrichum* were identified in five plant species. *Phyllosticata* was found in all the plants except in *T. arjuna. Pestalotiopsis* spp. was found only in *T. alata*, *T. chebula*, and *T. crenulated*, while *Xylaria* was present only in *T. chebula* and *T. crenulata*. *Sporomiella* and *Lasiodiplodia* were found only in *T. chebula* [52].

Endophytic fungi colonizing the inner bark and twigs of *T. arjuna* were investigated by Tejesvi et al. [51]. The results revealed a greater preference of fungi for bark than twigs, as exemplified by the colonization frequency of 18.5% for inner bark and 4.6% for twigs. Tejesvi et al. [59] also reported that the roots of *T. arjuna* were colonized by *Pestalotiopsis clavispora*, *Pestalotiopsis microspore,* and *Pestalotiopsis* sp., while the bark was mostly colonized by *Pestalotiopsis microspora*. Twenty endophytic fungi were isolated from the leaves, twigs, and bark tissues of the *T. arjuna* collected from Shirpur (India) and identified as *Aspergillus flavus, Diaporthe arengae*, *Alternaria* sp., and *Lasiodiplodia theobromae*, with *Aspergillus flavus* being the most predominant endophyte in the leaves and bark of the plant [45]. *Aspergillus flavus*, *Aspergillus niger*, and *Rhizophus oryzae* were reported as dominant endophytes from the leaves of *T. brownie* collected from the Ghindae sub zone in Eritrea [46]. In addition, *Aspergillus aculeatus*, *Aspergillus oryzae*, and *Curvularia* sp. were reported as endophytes of the leaves of *T. laxiflora* collected from Al-Zohriya Gardens in Giza, Egypt [47].

The endophytic fungi colonizing different organs of *T.catappa* and *T.mantaly* were investigated by Toghueo et al. [50]. A total of 111 fungi belonging to 14 different genera were obtained from *T. catappa* samples, with the highest infection frequency detected in stem fragments (86.7%) and the lowest in the root (5.0%). *Pestalotiopsis*, *Diaporthe*, and *Paraconiothyrium* were isolated only from stem samples; *Trichoderma*, *Botryosphaeria*, and *Lasiodiplodia* from the bark, and *Cladosporium*, *Corynescora,* and *Mycosphaerella* from leaf veins. Other taxa found in various plant parts were *Pseudocercospora*, *Guignardia*, *Ophioceras*, *Cercospora*, *Diaporthe*, *Pseudofusicoccum*, and *Xylaria*. For *Terminalia mantaly* samples, a total of 140 endophytes classified into 11 genera were isolated. *Pseudofusicoccum* and *Fusarium* were isolated only from the stem; *Cercospora*, *Septoria*, and *Phoma* were isolated from the leaf blade; *Corynespora*, *Nigrospora*, *Colletotrichum*, and *Cryptococcus flavescens* were isolated from the leaf vein; and *Lasiodiplodia* was isolated from the bark [50]. *Lasiodiplodia pseudotheobromae*, *Lasiodiplodia theobromae*, *Lasiodiplodia parva*, and *Endomelanconiopsis endophytica* were also isolated from the bark of *T. mantaly*, *T. ivorensis*, and *T. superba* growing in Cameroon [54].

This tissue-dependent specialization in host–endophyte relations is not unusual and has been evidenced in other plants genera. In fact, endophytes are often tissue specific, but most only show a tissue preference [70]. This may stem from a high affinity of endophytes to establish within a specific chemistry or texture of different host tissues [71]. Isolation methods used to study endophytes tissues colonization are mainly inaccurate. However, powerful image analyses can provide information about the exact endophyte colonization of plant tissues and about physical contacts between different microbial groups [72,73]. Most of the genera reported from these *Terminalia* spp. have belonged to the phylum Ascomycota, have been isolated as endophytes from many different plant taxa, and have been reported as dominant components in tropical trees [74,75,76]. Moreover, the fact that genera such as *Colletotrichum*, *Lasiodiplodia*, *Pestalotiopsis*, and *Phomopsis* are found in the tissues of almost all *Terminalia* spp. investigated so far may be related to their ability to colonize plant tissues which favor their establishment and growth [77].

## 3. Biological Activity of Endophytic Fungi from Terminalia Species

Endophytic fungi are increasingly recognized as a novel source of bioactive metabolites with therapeutic potential. It has been suggested that the close biological association between endophytes and their plant host results in the production of a greater number and diversity of biological molecules compared to epiphytes or soil-related microbes. Moreover, the symbiotic nature of this relationship indicates that bioactive compounds from endophytes are likely to possess reduced cell toxicity, as these chemicals do not kill the eukaryotic host system. This is of significance to the medical community, as potential drugs may not adversely affect human cells [78]. The secondary metabolites produced by endophytes associated with medicinal plants can be exploited for curing many diseases [79]. Over the past decade, the interest to explore endophytic fungi from the *Terminalia* genus as potential producers of biologically active products has increased, and many of these investigated so far have exhibited interesting activity (Table 2).

### 3.1. Antimicrobial Activities of Fungal Endophytes

The increasing number and worldwide distribution of resistant pathogens to antimicrobial drugs is potentially one of the greatest threats to global health, leading to health crises arising from infections that were once easy to treat. Infections resistant to antimicrobial treatment frequently result in longer hospital stays, higher medical costs, and increased mortality [93]. Therefore, new antimicrobial agents are in urgent demand for the treatment of microbial infections. This urgency has been the driving force behind the exploration of endophytes from *Terminalia* spp. as a source of natural products displaying broad spectrum of antimicrobial activities. The crude ethyl acetate extract of *Acremonium sclerotigenum* isolated from the leaves of *T. bellerica* showed potent antimicrobial activity against gram positive (*Bacillus cereus*, *Bacillus subtilis*, and *Staphylococcus aureus*) and gram negative (*Escherichia coli* and *Pseudomonas aeruginosa*) bacteria, with an inhibition diameter ranging from 3.5–10mm [43]. Extracts from two *Aspergillus* spp. isolated from the leaves of *T. brownii* exhibited antimicrobial activity against *Staphylococcus aureus* [46]. The ethyl acetate extracts of the *Pestalotiopsis* species endophytes of *T. arjuna* and *T. chebula* showed greater antifungal activity against *Alternaria carthami*, *Fusarium oxysporum*, *Fusarium verticilloides, Macrophomina phaseolina*, *Phoma sorghina*, and *Sclerotinia sclerotiorum*, with the zone of the inhibition diameter ranging from 4 to 25 mm [59] and antibacterial activity with >75% inhibition against five bacterial strains including *Bacillus subtilis*, *Escherichia coli*, *Pseudomonas fluorescens*, *Xanthomonas axonopodis pv*. *malvacearum*, and *Staphylococcus aureus* [83]. Endophytic fungi obtained from *T. arjuna* were screened against a panel of bacteria and fungi. The crude extracts showed considerable antimicrobial activity against common human bacterial (*Staphylococcus aureus, Escherichia coli, Pseudomonas aeruginosa, Proteus vulgaris, Salmonella abony*, and *Bacillus subtilis*) and fungal (*Candida albicans, Aspergillus niger*, and *Penicilium* sp.) pathogens, with inhibition diameters ranging from 16–32mm [45]. The antimicrobial activity exhibited by crude extracts from endophytic fungi may suggest that the further investigation of these potent extract could lead to the identification of metabolites with the potential to treat bacterial and fungal infections.

Indeed, the ethyl acetate extract prepared from the fermentation broth of *Alternaria alternate* Tche-153 isolated from *Terminalia chebula* was reported to exhibit significant ketoconazole-synergistic activity against *Candida albicans*. The bioassay directed fractionation of that potent extract led to the isolation of altenusin (**1**), isoochracinic acid (**2**), and altenuic acid (**3**) together with 2,5-dimethyl-7-hydroxychromone (**4**) (Figure 1). Moreover, altenusin, in combination with each of the three azole drugs—ketoconazole, fluconazole or itraconazole—at their low sub-inhibitory concentrations, exhibited potent synergistic activity against *C. albicans*, with a fractional inhibitory concentration index range of 0.078 to 0.188. This suggests that altenusin may be an azole-synergistic prototype for the further development of new synergistic antifungal drugs to treat invasive candidiasis [44]. Crude ethyl acetate extracts from *Trichoderma atroviride*, *Trichoderma* sp., *Penicillium chermesinum*, and *Aspergillus niger* isolated from *T. catappa*, as well as *Diaporthe* sp., *Phomopsis* sp., and *Cryptococcus flavescens*, isolated from *T. mantaly* were tested against a set of bacteria and fungi. An extract from *T. atroviride* was the most active against yeasts (MIC from 0.019 to 5 mg/mL) and *P. chermesinum* (MIC from 0.625 to 0.019 mg/mL) was against bacteria; the chemical analysis of these extracts revealed the presence of 8 and 12 compounds, respectively, amongst which genipinic acid (**5**), citreomantanin, dermoglaucin (**6**), alternariol-5-methylether (**7**), ascochitine (**8**), and aflatoxicol were identified [48].

*Muscodor albus*, an endophytic fungus from *T. prostrata*, was investigated for its ability to produce volatiles organic compounds with antimicrobial activity. The results showed that *M. albus* produced volatiles compounds including naphthalene (**9**), and naphthalene, 1,1-oxybis-, exhibiting a 100% inhibition of bacteria and fungi except *Bacillus subtilis* and *Saccharomyces cerevisiae* [84]. In addition, *Oidium* sp., recovered as an endophyte in *T. catappa*, produced volatile organic compounds (VOCs) consisting of esters of propanoic acid, 2-methyl- (**10**), butanoic acid, 2-methyl- (**11**), and butanoic acid, 3-methyl- (**12**) (Figure 1). Moreover, the exogenous addition of isobutyric acid and naphthalene, 1,10-oxybis caused a dramatic synergistic increase of the antibiotic activity of the VOCs of *Oidium* sp. against *Pythium ultimum* [57]. Crude metabolites produced by some endophytes from *Terminalia* spp. were also reported to inhibit several *Plasmodium* strains. In fact, 10 and 12 endophytic fungi, respectively, from *T. catappa* and *T. mantaly* were cultured in three different culture media (PDB, MEB, and CDB), and the resulting ethyl acetate extracts were investigated for the antiplasmodial activity against sensitive and resistant strains of *P. falciparum*. The results showed that, irrespective of the medium used, the individual EtOAc extracts of *Aspergillus niger* 58 (IC_50_ 2.54–6.69 μg/mL) from the leaves of *T. catappa*, *Phomopsis* sp. N114 (IC_50_ 0.34–7.26 μg/mL) and *Xylaria* sp. N120 (IC_50_ 2.69–6.77 μg/mL) from the leaves of *T.mantaly* displayed very good antiplasmodial potencies [89]. These findings may suggest that the further investigation of the metabolome of these fungi could lead to the identification of compounds that could work as starting points for drug discovery against a wide range of infectious diseases. 

*Pestalotiopsis virgatula*, an endophyte of *T. chebula*, was found to produce a large excess of a single metabolite, 9-hydroxybenzo[c]oxepin-3[1H]-one (**13**) when grown in the minimal M1D medium [88]. Interestingly, the analysis of the extract from a potato dextrose broth medium by the HPLC-PDA-MS-SPE-NMR hyphenated system led to the identification of a total of eight metabolites (Figure 1), including 9-hydroxybenzo[c]oxepin-3[1H]-one; (R)-3-hydroxy-1-[(R)-4-hydroxy-1,3-dihydroisobenzofuran-1-yl]butan-2-one (**14**); 1-(3,9-dihydroxy-1,3-dihydrobenzo[c]oxepin-3-yl) ethanone (**15**); (E)-2-(hydroxymethyl)-3-(4-hydroxypent-1-enyl) phenol (**16**); (R)-3-hydroxy-1-[(S)-4-hydroxy-1,3-dihydroisobenzofuran-1-yl]butan-2-one (**17**); cyclosordariolone (**18**) [1,6-dihydroxy-5-(hydroxymethyl)-1-methylnaphthalen-2(1H)-one]; pestalospirane A (**19**); and pestalospirane B (**20**); among these, six were new compounds [90].

### 3.2. Antioxidant Potential of Endophytes

In humans, oxidative damage and free radicals are associated with a number of diseases including atherosclerosis [94], Alzheimer’s disease [95], cancer [96], ocular disease [97], diabetes [98], rheumatoid arthritis [99], and motor neuron disease [100]. Antioxidants are therefore needed to prevent the oxidative stress-mediated toxicity caused by oxygen-free radicals [101]. The exploration of natural products has been an important source for antioxidants for decades. In this respect, crude ethyl acetate extracts of the *Pestalotiopsis* species endophytes of *T. arjuna* and *T. chebula* were found to demonstrate strong radical scavenging activity, with an IC_50_ ranged from 14 to 27 μg/mL, a strong inhibition of lipid peroxidation (IC_50_ ranged between 30 and 35.50 μg/mL), and antihypertensive activity (IC_50_ ranged between 21 to 37 μg/mL μg/mL) [83]. The crude ethyl acetate extract of twenty isolates of endophytic fungi from *T. arjuna* were tested for antioxidant and anti-inflammatory activities. The results showed that extracts of *Diaporthe arengae* exhibit DPPH scavenging activity, with an inhibition percentage of 69.56% [45]. The bioguided fractionation of that crude extract afforded semisolid phenolic compounds including benzene propionic acid, 3, 5-*bis* (1, 1-dimethylethyl)-4-hydroxy methyl ester (**21**); Pterin-6-carboxylic acid (**22**); and 2, 6-ditert-butyl-4- phenol (**23**) (Figure 2) with strong *in vitro* and *in vivo* anti-hypercholesterolemic activity in hRBC membranes through the inhibition of lipid peroxidation [82]. In addition, extracts of *Diaporthe* sp. from *T. mantaly* showed radical scavenging activity, with a percentage inhibition ranging from 9.93 to 41.13%. The HPLC-MS analysis revealed the presence of ten (10) compounds, including ascorbic acid (**24**), genipinic acid (**5**), 4-deoxybostrycin (**25**), bionectriamide B, and trisdechloronornidulin [48]. 

The ethyl acetate extracts of *Penicillium thiomii* isolated from *T. chebula* exhibited DPPH scavenging activity, with an IC_50_ value of 40.73 µg/mL, and the chemical investigation led to the isolation of a new cyclic heptanone, 4,7-dimethyl-1,3-dioxa-cyclohepta-2-one (**26**) [102]. From the same endophyte, Shoeb et al. [58] isolated another new compound named terminatone (**27**), along with three known compounds, ergosterol, 4-hydroxy benzaldehyde (**28**), and 4-hydroxy-hexadec-6-enoic acid methyl ester (**29**). *Pestalotiopsis microspore*, gathered from a stem of *T. morobensis*, was cultured in an MID medium, and the methylene chloride extract was found to exhibit antioxidant and antifungal activities. The bioguided fractionation led to the isolation of an isobenzofuranone, isopestacin [80], and pestacin [81] (Figure 2). Isopestacin (**30**) enacts antioxidant activity by scavenging superoxide and hydroxy free radicals in solution [80], while pestacin (**31**) exhibits antioxidant activity 11 times greater than trolox, a vitamin E derivative, primarily via the cleavage of an unusually reactive C-H bond and, to a lesser extent, O-H abstraction [81]. Few other isobenzofuranones have been isolated from sources such as fungi, liverworts, and higher plants [103,104,105,106,107], but isopestacin is the only one having a substituted benzene ring attached at the C-3 position of the furanone ring [80]. 

### 3.3. Anticancer Potential of Endophytes

The global cancer burden is estimated to have risen to 18.1 million new cases and 9.6 million deaths in 2018. One in five men and one in six women worldwide develop cancer during their lifetime, and one in eight men and one in 11 women die from the disease. Worldwide, the total number of people who are alive within five years of a cancer diagnosis, called the five-year prevalence, is estimated to be 43.8 million [108]. Many efforts need to be put into the search for potential inhibitors for cancer treatment. For the continued search for new anticancer compounds, the ethyl acetate extracts of *Penicillium thiomii* isolated from *Terminalia chebula* inhibited the growth of CaCo-2 colon cancer cell lines, with an IC_50_ of 44 μg/mL [85]. Tawfike et al. [47] successfully employed metabolomic tools to isolate secondary metabolites with activity against several cancer cells from an endophytic *Aspergillus aculeatus* isolated from the Egyptian medicinal plant *T. laxiflora*. The same methodology was employed by Tawfike et al. [53] to investigate extracts from *Curvularia* sp. exhibiting growth inhibition against a chronic myelogenous leukemia cell (K562). The metabolomic tools and dereplication methods using high-resolution electrospray ionization mass spectrometry directed the fractionation and isolation of *N*-acetylphenylalanine (**32**) and two linear peptide congeners of 1: Dipeptide *N*-acetylphenylalanyl-l-phenylalanine (**33**) and tripeptide *N*-acetylphenylalanyl-l-phenylalanyl-l-leucine (**34**) (Figure 3). The related natural product *N*-acetyl-l-phenylalanyl-l-phenylalaninol has been previously isolated from culture filtrates of the fungus *Emericellopsis salmosynnemata* by Argoudelis et al. [109,110].

The diterpenoid “taxol” (paclitaxel) has generated more attention and interest than any other new drug since its discovery due to its unique mode of action compared to other anticancer agents. In fact, paclitaxel act by reducing or interrupting the growth and spreading of cancer cells [111]. Taxol (**35**) is an approved drug for the treatment of advanced breast cancer, lung cancer, and refractory ovarian cancer. This outstanding anticancer drug was first isolated from *Taxus brevifolia* and other plants from the *Taxus* family, which are not only are rare and slow growing but produce small amounts of taxol [112]. Fortunately, the isolation and identification of the taxol-producing endophyte *Taxomyces andreanae* provided the world with an alternative approach to obtain a cheaper and more available product via microorganism fermentation [113]. This also brought to light an understanding that endophytic fungi from others medicinal plants can produce paclitaxel. In this respect, an endophytic fungus, *Pestalotiopsis terminaliae*, isolated from the fresh healthy leaves of *T. arjuna* was screened for the production of taxol in artificial culture medium. The results showed that *P. terminaliae* can produce 211.1 μg/litre of taxol (Figure 3). The fungal taxol extracted exhibited strong cytotoxic activity towards BT220, H116, Int 407, HL 251, and HLK 210 human cancer cells *in vitro* when tested using an apoptosis assay [86]. In addition, *Chaetomella raphigera* (strain TAC-15), isolated from the same tree, was grown in an MID liquid medium and found to produce 79.6 μg/L of taxol [87]. These findings are not isolated, because several endophytic fungi, including *P. pausiceta* isolated from *Cardiospermum helicacabum* [114], the *Pestalotiopsis microspora* endophyte isolated from *Taxodium distichum* [115], the *Phyllosticta spinarum* fungal endophyte found in *Platycladus orientalis* [116], and the *Bartalinia robillardoides* endophyte found in *Aegle marmelos* [117], were also reported to produce taxol in vitro. The awareness of the fact that fungal endophytes from *Terminalia* spp. can produce taxol may suggest that deeper biological and chemical explorations of the metabolome of these fungi may lead to the discovery of more potent drugs for the treatment of cancer.

### 3.4. *Other Biological Activities*

Apart from their ability to produce metabolites with antimicrobial, antioxidant, and anticancer activity, endophytic fungi from *Terminalia* spp. can also exhibit a biocontrol ability against plant pathogens. Indeed, two endophytic *Trichoderma* spp. from the bark of *T. catappa* were investigated for their ability to control the causative agent of common bean root rot (*Fusarium solani*). The results showed that *Trichoderma atroviridae* and *Trichoderma* sp. exert over 86% inhibition of *F. solani* growth (Figure 4). *T. atroviridae* also showed a high potential in promoting bean seed germination (100%) and in protecting bean seeds against the deleterious effects caused by *F. solani*, emphasizing its significant biocontrol potential *in vitro* [92]. Prathyusha et al. [43] also reported the potential of *Acremonium sclerotigenum* isolated from the leaves of *T. bellerica* to exhibit siderophore activity *in vitro*. 

Collection fungal endophytes from *T.catappa* and *T.mantaly* were also investigated for their ability to produce industrial enzymes. The results showed that isolates were able to produce amylase, cellulose, lipase, and laccase [50], among which *Penicillium* sp. 51 and *P. chermesimum* exhibited strong cellulase activity [91], demonstrating that these fungi could be useful for the production of enzymes of industrial importance. 

## 4. Brief Outline for the Isolation, Identification and Metabolic Profiling of Endophytic Fungi from *Terminalia* Species

The analysis of data revealed that cultivation-based techniques were the methods used for the isolation and testing of endophytic fungi from different *Terminalia* species. Plant materials required for the study were collected and transferred to the lab and sterilized using varied surface-sterilization protocols. The combination of ethanol (70–95%) with NaOCl (1–4%) as defined by Schulz et al. [118] was mostly used by authors, although the concentrations and exposure times differed from one study to another [49,51,54]. However, Patil et al. [45] reported the use of only 0.5% NaOCl for 2 min for the sterilization of leaves, bark and twigs. Meanwhile, only 70% isopropanol for 2 seconds was used by Tawfike et al. [53] for the sterilization of leaves. In almost all these studies, the protocol for sterilization was the same for all plant tissues investigated. In contrast, Toghueo et al. [50] applied different sterilization protocols depending on the sensitivity of plant organ. In fact, leaves and stems were sterilized using only 1% NaOCl for 10 min, and they were rinsed with sterile distilled water. Meanwhile, bark, root, and stem fragments were subsequently surface-sterilized using 70% ethanol (5 min), 1% NaOCl for 15 min, and 70% ethanol for 2 min; finally, they were rinsed with sterile distilled water.

Sterilized fragments were thereafter plated on an agar medium such as PDA, water agar, or malt agar supplemented with antibiotics such as streptomycin [49,51,54] or chloramphenicol [50,53]. Emerging fungi were purified and identified based on both their macroscopic and microscopic structures, as suggested in taxonomical classification guides [119,120]. The identities of fungi of interest were confirmed based on the analysis of nucleotide sequences of their internal transcribed spacer (ITS) regions of rDNA [121]. Sequences obtained were compared with sequences from available the GenBank using the BLAST program of the National Institute of Health [44,50].

For their metabolic profiling and biological activities, endophytes were cultured in either broth (YES, PDB, MEB, CDB, MID, CMC) or solid media (PDA, rice), and, after a required incubation period, cultures were extracted by maceration using solvents such as ethyl acetate and methylene chloride. The resulting crude extracts were subjected to various biological activities and cultures that were found to be positive in preliminary screens were taken up for the isolation of pure molecules. The chemical structures of these natural products were then elucidated using various analytical chemistry and spectroscopy techniques, as described by Amagata [122].

## 5. Conclusions and Perspectives

Our investigation shows that endophytic community in *Terminalia* species is highly diverse and influenced by the season, location, and host plant species. These fungal endophytes may be prospective producers of an abundant and dependable source of bioactive and chemically novel compounds for potential use in medicine, agriculture, and industry. In fact, this review has demonstrated that endophytes from *Terminalia* spp. may produce highly potent and widely acclaimed metabolites such taxol, isopestacin, and pestacin. Further, bioprospecting endophytes from *Terminalia* species may reveal more metabolites of value for therapy. 

Though *Terminalia* represents one of the most important genera of medicinal plants with diverse biological activity, it remains one of the most unexplored biomes in terms of the fungal endophyte community. In fact, our investigation shows that among the 200–250 *Terminalia* species reported, only 13 species have been investigated for their endophytic fungi content; more than 187 *Terminalia* spp. are yet to be explored for endophytes and their metabolites. Taking advantage of existing information and up-to-date technologies, a further exploration of endophytes from *Terminalia* species may well identify inhibitors capable of curbing infectious diseases and cancer. 

## Figures and Tables

**Figure 1 jof-05-00043-f001:**
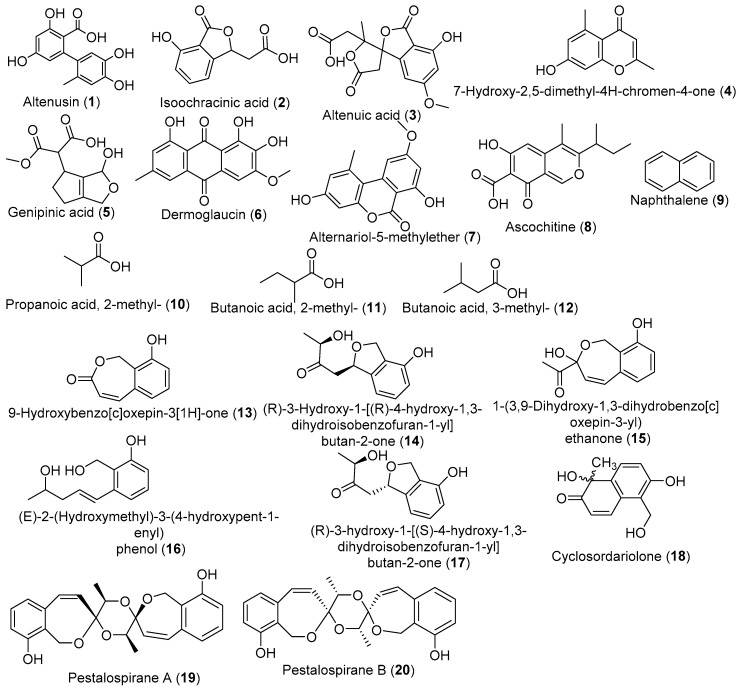
Diverse antimicrobial metabolites produced by endophytic fungi from the *Terminalia* species.

**Figure 2 jof-05-00043-f002:**
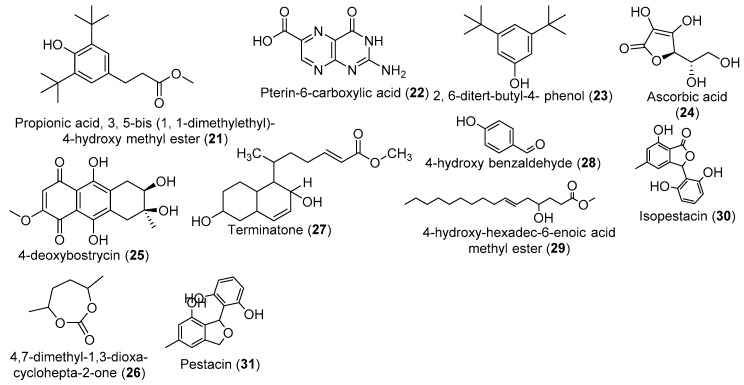
Metabolites isolated crude extracts with antioxidant activity.

**Figure 3 jof-05-00043-f003:**
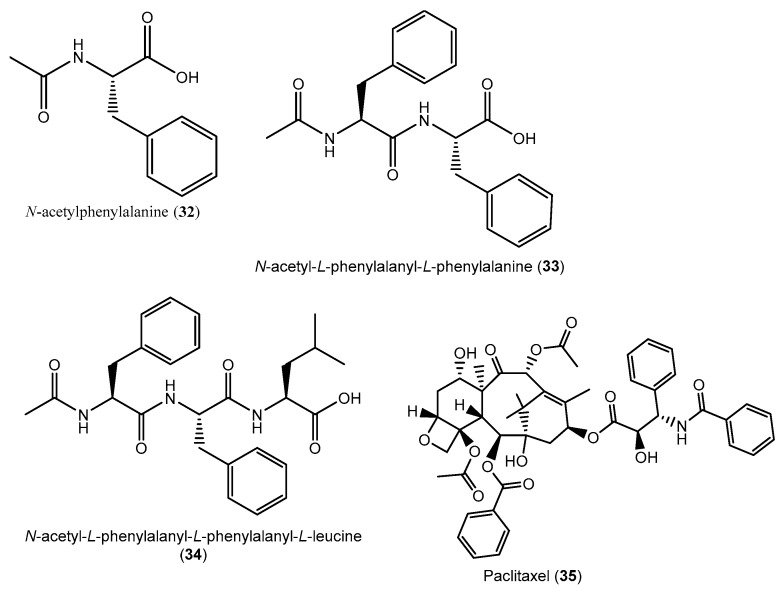
Metabolites isolated from endophytic fungi extracts exhibiting strong anticancer activity.

**Figure 4 jof-05-00043-f004:**
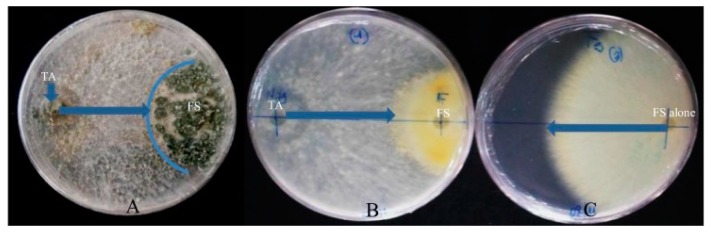
Antagonistic effects of *Trichoderma atroviridae* from *T.catappa* on *F. solani,* the causative agent of commom bean rot root. Diagonally opposed inocula were grown to assess the biological control potential of *Trichoderma* spp. over *F. solani*. Results indicated that the BCA grew faster and sporulated on the pathogen and, consequently, restricted its growth compared to the negative control [92].

**Table 1 jof-05-00043-t001:** Diversity of fungal genera isolated from *Terminalia* spp.

No.	Fungal Genus	Medicinal Plants	Organs of Isolation	Regions/Countries	Refs.
1	Acremonium	*T. bellerica*	Leaves	Telangana, India	[43]
2	Alternaria	*T. chebula*	Leaves	Bangkok, Thailand	[44]
		*T. arjuna*	Leaves, bark, twig	Shirpur, India	[45]
3	Aspergillus	*T. arjuna*	Leaves, bark, twig	Shirpur, India	[45]
		*T. brownie*	Leaves	Ghindae, Eritrea	[46]
		*T. laxiflora*	Leaves	Giza, Egypt	[47]
		*T. catappa*	Bark	Yaoundé, Cameroon	[48]
4	Botryodiplodia	*T. arjuna*	Bark	Srirangapatra, India	[49]
5	Botryosphaeria	*T. catappa*	Bark	Yaoundé, Cameroon	[50]
6	Cercospora	*T. catappa*	Leaves, stem, twig, bark	Yaoundé, Cameroon	[50]
		*T. mantaly*	Leaves		
7	Chaetomium	*T. arjuna*	Bark	Mysore region, India	[49]
			Bark, twig	Riparian, India	[51]
8	Chloridium	*T. arjuna*	Bark	Srirangapatra, India	[49]
9	Cladosporium	*T. arjuna*	Bark	Mysore region, India	[49]
		*T. catappa*	Leaves	Yaoundé, Cameroon	[50]
10	Cochlonema	*T. arjuna*	Bark	Mysore region, India	[49]
11	Colletotrichum	*T. alata*, *T. arjuna* ,*T. catappax*, *T. chebula* *T. crenulate*	Leaves	Kargudi, Ronohills, and Masinagudi, India	[52]
		*T. mantaly*	Leaves	Yaoundé, Cameroon	[50]
12	Corynescora	*T. catappa*	Leaves	Yaoundé, Cameroon	[50]
13	Cryptococcus	*T. mantaly*	Leaves	Yaoundé, Cameroon	[50]
14	Curvularia	*T. laxiflora*	Leaves	Giza, Egypt	[53]
15	Diaporthe	*T. arjuna*	Leaves, bark	Shirpur, India	[45]
		*T. catappa*	Stem	Yaoundé, Cameroon	[50]
		*T. mantaly*	Bark, leaves, stem	Yaoundé, Cameroon	
16	Endomelanconiopsis	*T. mantaly*, *T. ivorensis* *T. superba*	Bark	Belabo, Cameroon	[54]
17	Fusarium	*T. mantaly*	Stem	Yaounde, Cameroon	[50]
18	Gliocladium	*T. arjuna*	Bark	Nanjangud, India	[49]
19	Glomerella	*T. chebula*	Leaves	Tak province, Thailand,	[55]
20	Guignardia	*T. catappa*	Bark, leaves, stem	Yaoundé, Cameroon	[50]
21	Humicola	*T. arjuna*	Bark	Mysore region, India	[49]
22	Kwoniella	*T. bellerica*	Fruit	Kharagpur, India	[56]
23	Lasiodiplodia	*T. chebula*	Leaves	Kargudi, Ronohills, and Masinagudi, India	[52]
		*T. arjuna*	Leaves, bark	Shirpur, India	[45]
		*T. catappa*, *T. mantaly*	Bark	Yaounde, Cameroon	[50]
		*T. mantaly*, *T. ivorensis*, *T. superba*	Bark	Belabo, Cameroon	[54]
24	Memnoniella	*T. arjuna*	Bark	Nanjangud, India	[49]
25	Monocillium	*T. arjuna*	Bark	Srirangapatra, India	[49]
26	Mycosphaerella	*T. bellerica*	Fruit	Kharagpur, India	[56]
27	Myrothecium	*T. arjuna*	Bark, twig	Riparian, India	[51]
		*T. arjuna*	Bark	Mysore region, India	[49]
28	Nigrospora	*T. arjuna*	Bark	Mysore region, India	[49]
		*T. mantaly*	Leaves	Yaoundé, Cameroon	[50]
29	Oidium	*T. catappa*	-	Costa Rica	[57]
30	Ophioceras	*T. catappa*	Leaves, stem, twig, bark	Yaoundé, Cameroon	[50]
31	Paecilomyces	*T. chebula*	Leaves	Tak province, Thailand	[55]
32	Paraconiothyrium	*T. catappa*	Stem	Yaoundé, Cameroon	[50]
33	Penicillium	*T. chebula*	-	Dhaka, Bangladesh	[58]
34	Pestalotiopsis	*T. alata*, *T. chebula*, *T. crenulata*	Leaves	Kargudi, Ronohills, and Masinagudi, India	[52]
		*T. arjuna*	Bark, twig	Riparian, India	[51]
		*T. chebula*	Bark	Gopalaswamy hills, India	[49,59]
		*T. arjuna*	Bark	Nanjangud and Srirangapatra, India	[49]
		*T. catappa*	Stem	Yaoundé, Cameroon	[50]
		*T. morobensis*	Stem	Papua New Guinea	[60]
35	Phialophora	*T. arjuna*	Bark	Mysore region, India	[49]
36	Phoma	*T. mantaly*	Leaves	Yaoundé, Cameroon	[50]
37	Phomopsis	*T. alata*, *T. arjuna*, *T. catappa*, *T. chebula*, *T. crenulata*	Leaves	Kargudi, Ronohills, and Masinagudi, India	[52]
		*T. chebula*	Leaves	Tak province, Thailand	[55]
38	Phyllosticata	*T. alata*, *T. catappa*, *T. chebula*, *T. crenulata*	Leaves	Kargudi, Ronohills, and Masinagudi, India	[52]
39	Pseudocercospora	*T. catappa*	Leaves, stem, twig, bark	Yaoundé, Cameroon	[50]
40	Pseudofusicoccum	*T. mantaly* *T. catappa*	Stem	Yaoundé, Cameroon	[50]
41	Rhizophus	*T. brownie*	Leaves	Ghindae, Eritrea	[46]
42	Septoria	*T. mantaly*	Leaves	Yaoundé, Cameroon	[50]
43	Sporomiella	*T. chebula*	Leaves	Kargudi, Ronohills, and Masinagudi, India	[52]
44	Stemphylium	*T. arjuna*	Bark	Mysore region, India	[49]
45	Trichoderma	*T. arjuna*	Bark	Nanjangud and Srirangapatra, india	[49]
		*T. catappa*	Bark	Yaoundé, Cameroon	[50]
46	Tubercularia	*T. arjuna*	Bark, twig	Riparian, India	[49,51]
47	Xylaria	*T. chebula* *T. crenulata*	Leaves	Kargudi, Ronohills, and Masinagudi, India	[52]
		*T. catappa*	Leaves, stem, twig, bark	Yaoundé, Cameroon	[50]

**Table 2 jof-05-00043-t002:** Biological activities of fungal endophytes from different species of *Terminalia* plants.

Endophytic Fungal	Host Plant	Culture Conditions	Compound/Extract	Activities	References
*P. microspore*	*T. morobensis*	MID, 35 days, 23 °C (no shaking)	MeCl extract	Antimicrobial, Antioxidant, Antimalarial	[80]
			Isopestacin	Antioxidant,	[80]
			Pestacin	Antioxidant,	[81]
*D. arengae,*	*T. arjuna*	SBM, 28 °C, 16 days, shaking (150) rpm	EtOAc extract	Antimicrobial, Antioxidant,	[45]
			Benzene propionic acid, 3, 5–bis (1, 1-dimethylethyl)-4-hydroxy methyl ester; Pterin-6-carboxylic acid; 2, 6-ditert-butyl-4- phenol	Anti-inflammatory, Anti-hypercholesterolemic	[82]
*Pestalotiopsis* spp.	*T. arjuna*	PDB, 21 days, 23 °C (no shaking)	EtOAc extracts	Antifungal	[59]
				Antibacterial, Antioxidant, Antihypertensive	[83]
*Acremonium sclerotigenum*	*T. bellerica*	Rice, 25 °C, 30 days (No shaking)	EtOAc extract	Antimicrobial, Siderophore production	[43]
*Muscudor albus*	*T. prostrata*	PDA, 4 days, 23 °C (no shaking)	naphthalene, and naphthalene, 1,19-oxybis-	Antimicrobial	[84]
*Oidium* sp.	*T. catappa*	PDA,10 days, 23 °C (no shaking)	Volatile organic compounds	Antifungal	[57]
*Penicillium thiomii*	*T. chebula*	PDA, 21 days, 25 °C (no shaking)	EtOAc extract	Anticancer	[85]
			Terminatone	Antioxidant	[58]
*P. terminaliae,*	*T. arjuna*	MID, 21 days, 26 °C (no shaking)	Paclitaxel/ MeCl extract	Anticancer	[86]
*C. raphigera*	*T. arjuna*	MID, 21 days, 26 °C (no shaking)	Paclitaxel/MeCl extract	Anticancer	[87]
*Aspergillus* spp.	*T. brownii*	SDB, 9 days, 30 °C (no shaking)	-	Antimicrobial	[46]
*A. alternata*	*T. chebula*	YES, 21 days, 25 °C (no shaking)	EtOAc extract	Antifungal	[44]
			Altenusin		
			Isoochracinic acid		
			Altenuic acid		
			2,5-dimethyl-7-hydroxychromone		
*A. aculeatus*, *A. oryzae*	*T. laxiflora*	RM, (no shaking)	EtOAc extract	Anticancer	[47]
*Curvularia* sp.	*T. laxiflora*	RM, (no shaking)	EtOAc extract	Anticancer	[53]
*Pestalotiopsis virgatula*	*T. chebula*	M1D, 21 days, 23 °C (no shaking)	9-Hydroxybenzo[c]oxepin-3[1H]-one/ EtOAc extract	Antimicrobial	[88]
*Aspergillus niger*	*T.catappa*	PDB, MEB and CDB, 6 days, 25 °C (no shaking)	EtOAc extract	Antiplasmodial	[89]
*Phomopsis* sp. *Xylaria* sp.	*T.mantaly*				
*Pestalotiopsis virgatula*	*T. chebula*	PDB, 21 days, 23 °C (no shaking)	EtOAc extract	-	[90]
			Cyclosordariolone		
			(R)-3-Hydroxy-1-[(R)-4-hydroxy-1,3-dihydroisobenzofuran-1-yl]butan-2-one;		
			(R)-3-Hydroxy-1-[(S)-4-hydroxy-1,3-dihydroisobenzofuran-1-yl]butan-2-one;		
			(E)-2-(Hydroxymethyl)-3-(4-hydroxypent-1-enyl) phenol;		
			1-(3,9-Dihydroxy-1,3-dihydrobenzo[c]oxepin-3-yl) ethanone;		
			Pestalospirane A;		
			Pestalospirane B		
			9-Hydroxybenzo[c]oxepin-3[1H]-one		
*T. atroviride* *P. chermesinum*	*T.catappa*	RM, 40 days, 25 °C, (No shaking)	EtOAc extract	Antimicrobial	[48]
*Diaporthe* spp.	*T.mantaly*			Antioxidant	
*Penicilium* spp.	*T.catappa*	CMC, 6 days, 25 °C (No shaking)	*-*	Cellulase activity	[91]
*Trichoderma* spp.	*T.catappa*	PDA, 10 days, 25 °C (No shaking)	*-*	Biocontrol	[92]

Minimal M1D: D-glucose 70.0 mg, l-asparagine 2.0 mg, yeast extract 1.0 mg, KH_2_PO_4_ 1.5 mg, MgSO_4_·7H_2_O 0.5 mg, FeSO_4_·7H_2_O 1.0 mg, MnSO_4_·H_2_O 1.0 mg, CuSO_4_·5H_2_O 1.0 mg, ZnSO_4_·7H_2_O 1.0 mg, 1L distilled water); Soybean meal medium (SBM): Lactose, 60mg; soybean meal 25 g; KH_2_PO_4_ 2mg, K_2_HPO_4_ 1mg; MgSO_4_·7H_2_O 0.5mg, FeSO_4_·4H_2_O 1mg; NaNO_3_ 0.5mg, MnSO_4_·7H_2_O 0.5mg; Potato dextrose broth (PDB): potato infusion 200 g, dextrose 20 g, Distilled water: 1 liter, pH 5.1 ± 0.2; Rice Medium (RM): Rice 200g, distilled water 1L; Sabouraud dextrose broth (SDB): Dextrose 40 gm, Peptone 10 gm, Distilled Water 1000 mL; Yeast extract sucrose (YES): Yeast extract 4.000 gm/L, Sucrose 20.000 gm/L, Potassium dihydrogen phosphate 1.000 gm/L, Magnesium sulphate 0.500gm/L; malt extract broth (MEB): malt extract 17 g, mycological peptone 3 g, pH 5.4 ± 0.2; Czapek Dox (CDB): sucrose 30 g, sodium nitrate 2 g, dipotassium phosphate 1 g, magnesium sulphate 0.5 g, potassium chloride 0.5 g, ferrous sulphate 0.01 g, pH 7.3 ± 0.2.; CMC medium: Carboxymethyl cellulose medium; EtOAc: ethyl acetate; MeCl: methylene chloride.

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
