# Peer review of "Endophytic Fungi from Terminalia Species: A Comprehensive Review"

_jof, 2019, doi:10.3390/jof5020043_

Round 1

Reviewer 1 Report

The work collected articles related to endophytic fungi from Terminalia plants and sorted  the fungal diversities information according to some subjects such host plant species, location, season and tissues, furthermore, summing and stating some potential bioactive functions of their compounds.  The paper sounds to be somehow helpful  on endophytic fungi from the medicine plants in coming time. The current version needs a intensive revision on English grammar and term names formats. Some examples below

line 22, gives.

line 24, Terminalia.  Each genus and species name must be italic in text, tables and figures.Please check it through whole text.

line 32, bacteria and fungi.

line 34, diseases.

line 77, spp.   The word is not spp and must be upright. Please check it through text.

line 79, summarized.

line 90, were.

line 110, sp.  must be sp. not sp and upright.

lines 122-124, was, favo(u)red, aided, showed. Many such error tension in the text.

lines 148-150, This sentence needs to be improved.

line 160, fungi.

line 232, The first word were should be deleted.

line 242, The first letter of a compound common name is not necessary to be uppercase.

lines 262 and 263,  What's the 30 and 42?

line 366, deleting a "from" .

Author Response

The current version needs a intensive revision on English grammar and term names formats.

Response: the main manuscript have been reviewed. Your suggestions and comments were used to improve the content of the manuscript. Modification made are highlight in red in the main text.

Reviewer 2 Report

Your review seems a valuable guide to the ‘as yet’ shallow research into the potential of a large genus. The English expression needs careful revision.  The work quoted is impressive - but may be undervalued if you write too subjectively.  Remove subjective writing and the work will appear as the substance it represents. No need to overstate speculation (eg Conclusions).

ABSTRACT

The abstract is rather general and not sufficiently informative: Delete sentence 3.  Inform us as to the size of the Terminalia genus. Make the abstract attract full-text readers by mentioning a couple of important ‘established’ endophytes and their metabolites that are already in significant use. Similarly, mention a couple that appear to show promise for development.

INTRODUCTION

Mention history. When were endophytes first recorded? Was it Acremonium in Lolium multiflorum in Germany in the 1880s? That suggests that we have been slow to follow up!

Line 32  change to “bacteria and fungal species that colonise and speed…..”

L42  reasons, not reasoning

L47   produce

L67  all should be in past tense

BIODIVERSITY

Comment on the relative number of endophytes in Table 1 that produce spores as distinct from those that rely on asexual reproduction and colonization of developing seed.

L89  insert ‘specimens’ after arjuna. Say riparian locations in India

L92-3  delete ‘it is the case of’ and insert ‘was’ after Pestalotiopsis.

L95  insert ‘of endophytes’ after ‘diversity’ and delete ‘in endophytes composition’.

L113 and L160:      fungi, not fungal

L119  previous not previously

L123  insert ‘the’ before rainy

L133  which references?

L141  When rather than ‘in fact’

L143  delete full stop and ‘Overall’

L144  delete the second ‘the’

L145  delete the first ‘in’

L149  insert a semi-colon after arjuna, delete ‘with’. Insert ‘in’ before ‘twigs’

L150  delete either ‘as well’ or ‘also’

L152 and L232:    fungal not fungi

L154  what is MS?

L155  insert ‘the’ before leaves

L163-170  Are italics needed?

L174  plant, not plants

L177  try ‘endophyte colonization of plant tissues’.  This sentence needs expanding as to what you mean.

L188  reference needed here re the “suggestion”

L197  delete profiles

Table 2: re Oidium spp.,   change to volatile organic as you prefer elsewhere (e.g. L257). Ditto re L254 and L255

ANTIMICROBIAL ACTIVITIES

L236  ranging, not ranged

L242-3, 302:  separate words.  Do compounds require capitalization? I think not.

L253  “its”, not his

L254

SECTION 3.2

L300  3 decimal places inappropriate

L302  nivalenol is a food poison – a Fusarium mycotoxin

Fig 2 caption:  Is ‘Some’ a better word here rather than ‘few’?

L399  has, not have                                            

L348, 351  produce, not produced

L355  “findings”, not funding.  Delete ‘species’

CONCLUSION

L389  change ‘as a function of’ to ‘and influenced by season, location and host plant species’

L391  for potential use in med., ag. and ind.

2nd sentence needs rewording: Stick to facts.  Would “may be” better than “are”?

3rd sentence: delete first 6 words. Would “widely acclaimed” be better than “world class”?

There is no need to overdo emotional tone.

4th sentence: ‘further’ rather than ‘therefore’ ,  ‘may reveal more metabolites of value for therapy’

2nd paragraph: Although Terminalia represent…..     delete ‘the genus of’.  Insert ‘genera’ after ‘important’ sources of medicinal plants. 

L399  content; more than 187 Terminalia spp. are yet to be explored for endophytes and their metabolites.

L401  delete ‘deep’ and insert ‘further’

L402  last sentence:  ‘may well identify inhibitors capable of curbing infectious diseases and cancer’ rather than ‘can’. 

Author Response

1. Remove subjective writing and the work will appear as the substance it represents. No need to overstate speculation (eg Conclusions).

Response: Subjective writing and speculation have been removed

2. The abstract is rather general and not sufficiently informative: Delete sentence 3.  Inform us as to the size of the Terminalia genus. Make the abstract attract full-text readers by mentioning a couple of important ‘established’ endophytes and their metabolites that are already in significant use. Similarly, mention a couple that appear to show promise for development.

Response: Sentence 3 was deleted, the size of Terminalia genus was included as well as important metabolites isolated such as taxol and isopestacin.

3. Mention history. When were endophytes first recorded? Was it Acremonium in Lolium multiflorum in Germany in the 1880s? That suggests that we have been slow to follow up!

Response: Endophytes were first described by the German botanist Johann Heinrich Friedrich Link in 1809. However, Acremonium endophyte of Lolium multiflorum was reported in 1904 by Freeman. It has been 210 years since the discovery of endophytes. Although, several progress have been made in the understanding of plant-endophyte relationship as well as their potential, many are yet to be discovered.

In the present review, rather than mentioning history of endophytic fungi, we found more appropriate to fix the context of the study around the importance of microbial community within the host plant. We thought that this perspective will be more attractive for the subject been developed.

4. Comment on the relative number of endophytes in Table 1 that produce spores as distinct from those that rely on asexual reproduction and colonization of developing seed.

Response: My apology, I didn’t understood this particular suggestion.

5. For suggestions for english grammar and term names formats

Response: The main manuscript have been reviewed. Your suggestions and comments were used to improve the content of the manuscript. Modification made are highlight in red in the main text.

Reviewer 3 Report

I have reviewed the manuscript “Endophytic fungi from Terminalia species: A Comprehensive Review” by Toghueon & Boyom and have the following comments:

There needs to be  a comprehensive editing of the language.

Overall, the material presented should be organised in   better manner with more subheadings to direct the reader’s attention.

 What is lacking is a section on the methodology used to obtain h as it has a strong influence on the extent of diversity obtained. This should include surface sterilisation methods, media used for isolation, plating out of plant material, and   conditions and length (time) of incubation. One would like to know, for example, how endophytes were obtained from Bark as opposed to leaves. Were any selective agents used?

The comment on the 30,000 plant species line 36) needs clarification- do they mean one or more endophyte unique to the plant species? From the review that has been presented are they able to confirm this hypothesis? The other common statement is that medicinal plants are selected “since the medical uses to which the plant may have been selected relates more to its population of endophytes than to the plant biochemistry itself” (lines 44-50).

In the fungi under review has this hypothesis also been proven?

 Table 1 is a good example of getting organisation into the review and I would recommend that more tables (and figures) be used especially for Section 2.1. This section repeats what is presented in Table 1.  The points on the effects of climate, location can be discussed without repeating what is in the table. Can the information on seasons (line 120 onwards) be placed in a table?

The same applies to section 2.2 as a the reported fungal names are difficult to follow when they are in free text-  a figure of a tree with location of fungi isolated from the different parts may be a more efficient method of conveying the same information.

 Section 3.1 is mostly covered in Table 2 and is very repetitive and dry.

It appears that the chemical structures are included without explaining the context and then title “Brief look..” is exactly what it should not be. Each structure should be numbered with  reference to the producing organism. The section 3.2 on antioxidant activity of extracts should be on pure compounds rather than of solvent extracts. The chemical structures if shown need to be “comprehensive”as in the title of the manuscript and not “Few antioxidant metabolites…”.

The taxol story is very interesting and suggests that its production may be broader within fungal endophytes.

Finally I do not understand why a plate (Fig 4) displaying antifungal activity is shown at the end under “other biological activities”

Author Response

1. There needs to be  a comprehensive editing of the language.

Response: The main manuscript have been reviewed. Your suggestions and comments were used to improve the content of the manuscript. Modification made are highlight in red in the main text.

2.What is lacking is a section on the methodology used to obtain h as it has a strong influence on the extent of diversity obtained. This should include surface sterilisation methods, media used for isolation, plating out of plant material, and   conditions and length (time) of incubation. One would like to know, for example, how endophytes were obtained from Bark as opposed to leaves. Were any selective agents used?

Response: the summary and discussion of methods for fungi isolation, identification and metabolic profiling have been included in section 4.

The comment on the 30,000 plant species line 36) needs clarification- do they mean one or more endophyte unique to the plant species? From the review that has been presented are they able to confirm this hypothesis? The other common statement is that medicinal plants are selected “since the medical uses to which the plant may have been selected relates more to its population of endophytes than to the plant biochemistry itself” (lines 44-50).

 Response: The sentence (Of note, of the nearly 300,000 plant species forming the vegetal biodiversity of earth, each individual plant is host to one or more endophytes) simply means that each of the plant can be the host of one or more endophytic fungi species. Also the claim that the medical uses to which the plant may have been selected relates more to its population of endophytes than to the plant biochemistry itself is supported by Reviews by Khare et al. 2018 (Front. Microbiol); Venieraki et al. 2017 (Hellenic Plant Protection Journal); Jia et al. 2016 (Front. Microbiol) and Hardoim et al. 2015 (Microbiol. Mol. Biol. Rev.).

Table 1 is a good example of getting organisation into the review and I would recommend that more tables (and figures) be used especially for Section 2.1. This section repeats what is presented in Table 1.  The points on the effects of climate, location can be discussed without repeating what is in the table. Can the information on seasons (line 120 onwards) be placed in a table?

Response: The section 2.1 is the interpretation and analysis of data reported in table 1 reason why it seems repetitive. We thought that including these informations in the table will facilitated the comprehension of the analysis made in the main text. Unfortunately, information regarding seasons have been mantion only in very very few articles. Thus we didn’t find necessary to include in the table.

It appears that the chemical structures are included without explaining the context and then title “Brief look..” is exactly what it should not be. Each structure should be numbered with reference to the producing organism.

Response: The title of figure have been changed as suggested. Also, each of the chemical structured presented is cited in the main text with to the producing organism. Therefore, repeating the name of producing fungi in the figure could increase repetition in the document.

 The section 3.2 on antioxidant activity of extracts should be on pure compounds rather than of solvent extracts. The chemical structures if shown need to be “comprehensive”as in the title of the manuscript and not “Few antioxidant metabolites…”.

Response: Giving the fact that our goal was to present all reported activity regarding the potential of endophytic fungi from Terminalia, we found necessary to mention not only the activity of compounds but also for crude extracts even though compounds have not been isolated from some of these extracts.  

The taxol story is very interesting and suggests that its production may be broader within fungal endophytes.

Response: Yes, taxol production may be broader within fungal endophytes because more and taxol-producing fungal are been reported (Li et al. 2009; Elavarasi et al 2012; Roopa et al. 2015).

Finally I do not understand why a plate (Fig 4) displaying antifungal activity is shown at the end under “other biological activities”

Response: To investigate the biological control potential of a given organism, the control agent is often tested in petri plate against the pathogenic organism using the dual culture method. Only biocontrol agents able to significantly inhibit the growth of the pathogen in this preliminary assay are progress further. The plate in figure 4 is displaying the potential of Trichoderma atroviridae to control Fusarium solani in dual culture plate. This method was used to identify this endophytic fungi as potential biocontrol agent.   

Round 2

Reviewer 1 Report

The revised version has been improved some issues I concerned for the manuscript.

Author Response

Figures in the manuscript have been reedited

Reviewer 3 Report

The authors have addressed most of my concerns but not all. The surface sterilisation methods should come earlier in the review as readers would like to find out whether the methods change when treating different tissues.

I still have a problem with the lack of a link of the chemical structures to the text. Here again     my suggestion that numbers are used has been ignored.

Otherwise I find that the amendments made have improved the text considerably.    

Author Response

The surface sterilisation methods should come earlier in the review as readers would like to find out whether the methods change when treating different tissues.

Answer: Dear reviewer, thanks very much for your very useful comments that help improve the content of the manuscript. 

The reason why we choose to present the brief summary of methods used to study Terminalia's endophytes at the end of the manuscript it because we believe it to be a result that came out from our investigation.